# Government Expenditure on Maternal Health and Family Planning Services for Adolescents in Mexico, 2003–2015

**DOI:** 10.3390/ijerph17093097

**Published:** 2020-04-29

**Authors:** Leticia Avila-Burgos, Julio César Montañez-Hernández, Lucero Cahuana-Hurtado, Aremis Villalobos, Patricia Hernández-Peña, Ileana Heredia-Pi

**Affiliations:** 1Center for Health Systems Research, The National Institute of Public Health, Cuernavaca 62100, Morelos, Mexico; leticia.avila@insp.mx (L.A.-B.); ileana.heredia@insp.mx (I.H.-P.); 2School of Public Health and Administration, Universidad Peruana Cayetano Heredia, Lima 15102, Peru; cahuana.luce@gmail.com; 3Center for Population Health Research, The National Institute of Public Health, Cuernavaca 62100, Morelos, Mexico; alvillalobos@insp.mx; 4Netherlands Interdisciplinary Demographic Institute-KNAW, University of Groningen, 2511 CV The Hague, The Netherlands; hernandez@nidi.nl

**Keywords:** government health expenditure, adolescents, maternal health, family planning

## Abstract

The purpose of this study was to assess whether government policies to expand the coverage of maternal health and family planning (MHFP) services were benefiting the adolescents in need. To this end, we estimated government MHFP expenditure for 10- to 19-year-old adolescents without social security (SS) coverage between 2003 and 2015. We evaluated its evolution and distribution nationally and sub-nationally by level of marginalization, as well as its relationship with demand indicators. Using Jointpoint regressions, we estimated the average annual percent change (AAPC) nationally and among states. Expenditure for adolescents without SS coverage registered 15% for AAPC for the period 2003–2011 and was stable for the remaining years, with 88% of spending allocated to maternal health. Growth in MHFP expenditure reduced the ratio of spending by 13% among groups of states with greater/lesser marginalization; nonetheless, the poorest states continued to show the lowest levels of expenditure. Although adolescents without SS coverage benefited from greater MHFP expenditure as a consequence of health policies directed at achieving universal health coverage, gaps persisted in its distribution among states, since those with similar demand indicators exhibited different levels of expenditure. Further actions are required to improve resource allocation to disadvantaged states and to reinforce the use of FP services by adolescents.

## 1. Introduction

Mexico and other Latin American countries need to invest in the sexual and reproductive health of adolescents [1,2,3,4]. Ensuring the availability of healthcare and promoting healthy behaviors in this population group generate economic benefits that improve future labor productivity. Efforts on behalf of adolescents also contribute to reducing risks, such as complications during pregnancy in very young women, as well as preventing premature births and low-birthweight babies [3,5]. From a human rights perspective, adolescents are entitled to receive the information and health services they need to survive, grow, and realize their full potential as individuals [1,3,5].

As a result of Mexican population dynamics, adolescents now exert an unprecedented impact on national demographics. With nearly one-fourth of the population being aged 10–19 years in 2014 [6], the health needs of adolescents have acquired particular relevance. Their behavior has evolved towards early initiation of sexual activity [7,8] with limited use of contraceptive methods (CMs), especially among the poorer groups [9,10]. With a fertility rate that grew from 64 births per 1000 adolescents (15–19 years old) in 2009 to 77 births in 2014 [10,11], Mexico ranks first in adolescent fertility among Organization for Economic Cooperation and Development (OECD) member states [12]. The healthcare system needs to strengthen its response such that the health expectations and requirements of this age group are addressed and their access to health services is expanded. 

Over two decades ago, Mexico launched its System of Social Protection in Health (*Sistema de Protección Social en Salud*) with the aim of achieving universal health coverage. Its principal component, the *Seguro Popular* health insurance scheme, was conceived as a mechanism for enhancing service coverage and providing financial protection to the 45% non-salaried population in Mexico without access to social security (SS) [13,14]. Within this scheme, improving maternal health (MH) through greater coverage has been considered a core objective. An example of this commitment was the introduction of the Healthy Pregnancy Strategy (*Estrategia Embarazo Saludable*) in 2008. This program incorporated all pregnant women without SS coverage into the *Seguro Popular,* providing them with a package of MH services free of charge [15]. Although adolescents without SS coverage were not explicitly targeted by these initiatives, they benefited from the increased supply of maternal health (MH) and family planning (FP) services [16,17]. 

In 2009, Mexico stepped up its efforts to improve availability of sexual and reproductive health services for adolescents. One upshot was the Model of Comprehensive Care for the Sexual and Reproductive Health of Adolescents implemented in 2013. Two years later, having recognized adolescent pregnancy as a national problem, Mexico established the National Strategy for the Prevention of Adolescent Pregnancy. This initiative was designed to reduce the fertility rate among adolescents aged 15–19 years and to eradicate pregnancy among those aged 10–14 years [8]. One of its principal components was greater coverage for sexual and reproductive health services and modern contraception methods (CMs). To this end, the National Strategy engaged numerous public health institutions—those pertaining to the Social Security Institute, which covered 43% of the population, and those serving the population without SS coverage through the *Seguro Popular*, the State Health Services, and the *IMSS PROSPERA* Program, renamed *IMSS Bienestar* at the close of 2018 [18,19].

These initiatives expanded the coverage of antenatal care services [16] and improved the availability of CMs for the adolescent population lacking SS coverage [7,9], thereby mobilizing greater resources from the government. To facilitate the transfer of funds to State Health Services, the *System of Social Protection in Health* introduced a per capita payment scheme based on the number of affiliates. This was intended to overcome the historical inertia in the distribution of funds [20].

The Mexican *Reproductive Health Account* system offers information on public maternal health and family planning (MHFP) spending from 2003 to 2015 [21], which serves to monitor and analyze the resources and performance of these services during this period [17,19,22,23]. However, data are not disaggregated by age group, and the consequences of changes in MHFP spending for adolescents without SS coverage are therefore unknown. To assess whether policies for expanding the coverage of maternal health and family planning (MHFP) are benefiting the adolescents in need, we (1) estimated MHFP expenditure for the entire adolescent population without SS coverage and determined its growth and distribution among MH and FP services and providers for the period 2003–2015; (2) analyzed the expenditure gaps among states by level of marginalization; and (3) examined the relationship between MHFP spending and demand indicators. Given that changes in the health system have centered on the population without SS coverage, we confined our analyses to this population.

## 2. Material and Methods

We conducted an ecological study in order to estimate government MHFP expenditure for adolescents without SS coverage. To achieve this, we drew data for 2003–2015 from the *Reproductive Health Accounts* [21], constructed according to the *OECD System of Health Accounts* [24] and the *World Health Organization’s Guide to Producing Reproductive Health Subaccounts* [25].

### 2.1. MHFP Expenditure for Adolescents

Analysis included government strategies aimed at the population without SS coverage, namely the *Seguro Popular* insurance and care providers, the State Health Services, and the *IMSS PROSPERA* Program [19,21]. Expenditure for adolescents was grouped according to care providers, specifically hospitals and ambulatory-care centers [21,24]. The selection of beneficiaries focused on young people between 10 and 19 years old, the age range for adolescence established by the Specific Action Program for the Sexual and Reproductive Health of Adolescents [26]. 

To estimate expenditure for the adolescent population, we carried out the following procedures. First, we identified the number of MH and FP consultations offered by ambulatory-care centers to 10- to 19-year-olds without SS coverage. Calculations were undertaken by type of service (MH or FP), year, and state. Subsequently, we estimated the proportions of these consultations with respect to the total number of MH and FP consultations offered to the total population without SS coverage. Likewise, we weighted MH and FP expenditures incurred by hospitals using the proportions of in-hospital days and consultations offered to adolescents without SS coverage. Based on OECD and WHO methodology [24], we used these proportions to weight the MH and FP expenditures sustained by ambulatory-care centers and hospitals for the total population lacking SS coverage. The underlying assumption was that spending was similar for comparable types of in-hospital days and consultations regardless of the ages of patients. Analyses covered 131.9 million in-hospital days as well as 1571 million general and specialized in-hospital consultations. We obtained the data from the General Directorate of Health Information under the Federal Ministry of Health (*DGIS*) [27], and grouped them according to the International Classification of Diseases, 10th Revision (ICD-10) [28].

MH services included care during pregnancy, childbirth (vaginal or cesarean), and the postpartum period, as well as abortion. FP services included counseling, consultations, the provision of CMs (hormonal or otherwise), and the performance of definitive surgical procedures (tubal ligation and vasectomy) for the entire adolescent population, 10 to 19 years old. Consultations were grouped by provider (hospital or ambulatory-care center). 

We calculated MHFP expenditure in constant US dollars and converted the figures to 2015 international dollars (Purchasing Power Parity, PPP, 2015 US$1 = MXN 8.541) [29]. To establish a comparison among states, we adjusted MHFP expenditure by adolescent women aged 10–19 according to *DGIS* data [27].

### 2.2. Demand Indicators

Demand has been defined as the proportion of a population experiencing health needs and requiring health services to satisfy them. For the purposes of this study, we defined the demand indicator for MH services as the number of pregnant adolescents without SS coverage, and for FP services as the number of sexually active adolescent women without SS coverage. To estimate these indicators, we used data from the 2009 and 2014 National Surveys of Demographic Dynamics (*ENADID*s) [30]. Although the *ENADID* design identifies fertility and pregnancy at the population level, it captures information only on pregnancies among 15- to 54-year-old women. Thus, our demand indicator for MH services included only 15- to 19-year-old adolescents who were pregnant at the time of the surveys or the year before. Under the demand indicator for FP services, we included adolescent women between the ages of 15 and 19 who were sexually active during the month prior to the surveys and those aged 10 to 19 who became sexually active during the survey period.

To ensure comparability, we constructed the MHFP indicators for adolescents in accordance with international practices. For instance, we considered women as the basis of our CM demand indicators given the distinct impact that the use/non-use of CMs exerts on their reproductive health and risk of pregnancy. Moreover, female contraceptives offer a wider range of methods and costs compared to male contraceptives, limited mostly to condoms available at lower costs [31,32]. For these reasons, most resource-tracking methodologies, including the Health Accounts, base their indicators of FP spending on women as the common denominator [25]. Finally, international recommendations on investment in adolescent FP activities also consider women as the basis, particularly in light of the high-priority problem of adolescent motherhood [33]. We obtained data on the size of the adolescent population lacking SS coverage for the period 2010–2018 from the *DGIS* databases [27]. As information was unavailable for the period prior to 2010, we calculated data from 2003 to 2009 based on the average annual growth of the adolescent population without SS coverage during 2010–2018. States were grouped according to the State Marginalization Index of the National Population Council [34]. 

### 2.3. Analytical Strategy

To assess growth in MHFP spending during the period 2003–2015, we estimated the average annual percent change (AAPC) in MHFP expenditure through Jointpoint regression models [35]. Given the inertial allocation in the health budget [36], we adjusted the models using the logarithm of expenditure with autocorrelated errors. For 2009 and 2011, we inserted nodes on the introduction of the Healthy Pregnancy Strategy and its efforts to enroll the entire non-SS population in the *SP*, respectively. For each program, we estimated the ambulatory-care center/hospital ratio as an indicator of the relative growth in expenditure at ambulatory- and primary-care centers.

To analyze the alignment of expenditure with the population requiring MH and FP services among states, we calculated and assessed the concentration indices (CIs) for 2009 and 2014 based on their concentration curves. This involved the following procedures: (a) we calculated the proportions of public MH and FP expenditures spent in state j with respect to total MH and FP expenditures at the national level, respectively; (b) we calculated the proportion of the population requiring these services in state j with respect to the total population requiring these services at the national level; and (c) we generated concentration curves by arranging the states on the x-axis according to the proportions of populations needing these services, from the lowest to the highest, and connecting them with their respective proportions of expenditures on the y-axis. The CI is the area between the concentration curve and the diagonal, and its values range from −1 to 1, with zero denoting equality [37]. Finally, we analyzed the expenditure and population data for 2014 by creating scatter plots and estimating Spearman correlations to ascertain their relationships and statistical significance. We used STATA version 13.0 for the analyses [38]. 

### 2.4. Ethical Considerations

To obtain data on service production (in-hospital days as well as consultations offered at hospitals and ambulatory-care centers), we used secondary public sources and the *ENADID* database, neither of which contained personally identifiable information. For expenditures, we used the *Reproductive Health Accounts*. This study was approved by the Research Ethics Committee of the National Institute of Public Health (No. 577-2016).

## 3. Results

In 2015, MHFP expenditure for the adolescent population without SS coverage totaled US$428 million, 88% of which was spent on MH. Meanwhile, the AAPC for the period 2003–2011 amounted to 15.4% (CI95%: 14.3−16.5) (Figure 1b and Table 1). The MH/FP ratios stood at 21.3 in 2003 and 7.5 in 2015. At the national level, expenditure per adolescent woman in the 10–19 age group rose from US$17 in 2003 to $64 in 2015 (Figure 1a,b and Table 1).

MH and FP services demonstrated different rates of growth for expenditures, which modified the distribution of funds among healthcare providers (Figure 2). From 2003 to 2011, spending on antenatal care in ambulatory-care centers showed an AAPC of 29.3% (CI95%: 23.5–35.4). Accordingly, while 61% of expenditure in ambulatory-care centers was used to finance antenatal care in 2003, this figure had grown to 85% by 2015. Spending on postpartum care grew at an annual rate of 9.3% from 2003 to 2009 (CI95%: 5.4–7.8) and 5.5% during the rest of the period analyzed (CI95%: 4.2–6.8) (Table 1 and Figure 2a).

Hospital spending on MH complications and childbirth registered an AAPC of 14.3% (CI95%: 13.2–15.5 and 11.9–16.8, respectively) from 2003 to 2009; the AAPC for the rest of the period analyzed was lower. Meanwhile, hospital spending on antenatal consultations, at a lower level, showed an annual growth rate of 8.1% (CI95%: 4.0–12.5) from 2003 to 2009, and 3.0% (CI95%: 1.5–4.5) from that year until 2015 (Table 1 and Figure 2b). The ambulatory-care center/hospital ratio of expenditure on MH services reflected a higher growth rate in ambulatory-care centers—in 2003, for each dollar spent by hospitals on MH, ambulatory-care centers spent $0.3. By 2015, the ratio was 0.6. 

In contrast with MH, FP expenditure was predominantly spent at ambulatory-care centers (Figure 3). Until 2008, the trends and levels of FP expenditure at hospitals and ambulatory-care centers were similar. However, from 2009 to 2011, spending by ambulatory-care centers experienced its period of greatest growth, with an AAPC of 130.6% (CI95%: 25.5–323.7). Thus, although hospital spending grew at an annual rate of 6.6% (CI95%: 3.7–9.5), by the end of the period analyzed, the ambulatory-care center/hospital spending ratio for FP services was 11 (Figure 3 and Table 1).

Table 2 shows that national expenditure on MHFP per adolescent woman had an AAPC of 11.8% (CI95%: 9.2–14.5) during the period 2003–2015. The increase in spending per adolescent woman reduced the gap between states with higher and lower levels of spending by 48% (from 5.96 in 2003 to 3.15 in 2015). Furthermore, the expenditure gap between states with higher and lower levels of social marginalization diminished by 13% (1.92 in 2003 and 1.68 in 2015). Although regions classified as having moderate, high, and very high levels of marginalization increased their spending per adolescent by 12% per year, no significant differences emerged in the annual rates among the five regions.

Finally, Figure 4 shows the concentration curve for expenditures on MH and FP and the correlations between expenditures and their respective demand indicators. In 2009 and 2014, half of the states (n = 16) concentrated less than 30% of the expenditures (concentration index = 0.32 for both years) (Figure 4a). However, for FP expenditure, the levels of inequality increased between 2009 and 2014 (CI = 0.30 and CI = 0.38, respectively) (Figure 4b). On the other hand, although a positive and significant correlation emerged between expenditure on MH at the state level and its demand indicator (rho > 0.9476, *p* < 0.05), differences were observed in the distribution of spending. For example, states such as Michoacan (MICH) and Guerrero (GRO) exhibited different levels of spending (Figure 4c) despite having comparable proportions of pregnant adolescents. The situation was similar for expenditure on FP (rho > 0.9016, *p* < 0.05), where Guerrero (GRO) and Mexico City (CDMX) presented comparable proportions of sexually active adolescents (Figure 4d) but different expenditure levels. Likewise, Figure 4c indicates that the MH expenditure levels in states with a low marginalization status, such as Jalisco (JAL) were similar to those of highly marginalized states and smaller populations, such as Tabasco (TAB) and San Luis Potosí (SLP).

## 4. Discussion

The adolescent population is one of the groups where public policy could have the most dramatic impact given the repercussions for health and wellbeing and the economic benefits these policies would generate [1,4]. In a context of limited resources, it is crucial to identify the areas of opportunity in which investments in the healthy development of this population group would be most efficiently used. In this regard, the results of this study indicate that efforts by the Mexican government to expand healthcare coverage for the population without Social Security (SS) coverage benefited the non-SS adolescent population. Thus, government expenditure on maternal health and family planning (MHFP) showed an average annual percent increase of 15% for the period 2003–2011 but no significant change in the remaining years. During 2003–2015, MHFP expenditure for non-SS adolescents represented 25% of government MHFP expenditure for the total population lacking SS coverage [21]. Nonetheless, despite the growth in spending, inequalities in distribution persisted. The adolescent population living in the most marginalized states and suffering from the greatest level of economic inequality, as well as historical differences in resource allocation [16,17,39] continues to demonstrate below-average levels of per capita spending.

In terms of sexual and reproductive health, it has been documented that the Mexican adolescent population has low rates of contraceptive use and difficulty in planning their sexual lives [7,9,40]. For these reasons, the first contact this population has with health services is generally to receive antenatal care, and their use of CMs frequently begins after the first pregnancy [8]. These patterns result in a situation in which antenatal, childbirth, and postpartum care claim 88% of MHFP expenditure for adolescents. The results show a greater increase in expenditure for MH services in ambulatory-care centers than in hospitals. This difference is explained primarily by a surge in the volume of antenatal consultations offered to adolescents without SS coverage (which rose from 66,436 in 2003 to 1.56 million in 2015) [21]. Despite an improvement in antenatal coverage for pregnant adolescents rising from 61% to 71.8% between 2000 and 2012 [16], the figures continue to be lower, especially in marginalized communities, than those achieved for pregnant adults (20 years and older) [16,41,42]. Because pregnancy during adolescence increases the risk of obstetric complications [1], it is hardly surprising that a third of MH expenditure was spent on hospital care for complications arising during pregnancy, childbirth, and the postpartum period.

With regard to FP expenditure, the observed annual increase of 130% incurred by ambulatory-care centers from 2009 to 2011 was a reflection of various events: (a) an increase in the number of people enrolled in the *Seguro Popular* from 9.1 million in 2009 to 43.5 million in 2010, which enhanced the coverage of health services, FP included, for the population lacking SS coverage [9,13,14,43]; (b) the strengthening of the FP program specifically for adolescents, leading to an increase in post-obstetric-event contraception [42] and which boosted the percentages of sexually active adolescents using a CM in their first sexual intercourse from 43% to 66% in women and from 70% to 85% in men, where the latter was attributed to greater male condom use [44]; and (c) the centralization and increase in the purchase of CMs [8]. As a result of higher FP expenditure, spending per adolescent woman 10–19 years old rose from US$0.8 in 2003 to $7.5 in 2015, reaching the threshold of $2.93 per woman per year since 2011, as recommended by the Guttmacher Institute [33]. In spite of these advances, however, deficiencies in CM coverage persist, and sexually active adolescents continue to be the group with the lowest rate of CM use in Mexico [9,40]. Evidence indicates that investment in FP programs generates significant monetary and social returns [34,45]. It has been estimated that each additional dollar invested in satisfying the demand for modern CMs generates savings of US$2–6 in healthcare spending for pregnant women and newborns [34]. If we also include the long-term effects of reducing maternal and infant mortality and increased economic growth, savings rise to US$60–120 per dollar spent on FP services [45]. This underlines the relevance of continuing to invest in FP for adolescents, as well as the need to make more efficient use of these resources.

Our results document the growing importance of ambulatory-care centers as health-service providers. These providers will no doubt continue to increase in importance, given that the current health reform in Mexico, which has replaced the *Seguro Popular* with the Health Institute for Wellbeing, is oriented towards strengthening the primary-care model for the population lacking SS coverage [46,47]. Nonetheless, the heterogeneity that characterizes ambulatory-care providers in terms of resources and the quality of services delivered [16,48,49] suggests that it will be necessary to improve their capacity to respond, particularly with regard to their supply of modern CMs [49].

Although government MHFP spending has increased across the board nationally, discrepancies persist in its distribution among states, since those with similar adolescent populations in need of MH and FP services exhibit different levels of expenditure. This could be the result of various factors, such as the persistence of inertial allocations: (a) in 2015, 33% of the *Seguro Popular* budget earmarked for payroll continued to be allocated to State Health Services on the basis of long-established, routine procedures [39]; (b) the concentration of infrastructure and personnel have traditionally privileged some states at the expense of others; and (c) managerial capacities have diverged and continue to vary widely among State Health Services [17,48,50].

Our study had the following limitations: (1) our analysis was restricted to government schemes providing coverage for the population without SS coverage, which prevents generalizing results to the entire public health system. However, these schemes provided coverage to 45% of the Mexican population [13,14], and their expenditures represented 46% of total public health spending in 2016 [39]; and (2) using production data to estimate the distribution of expenditure among health conditions and/or diseases, which could lead to estimation errors. Nonetheless, the OECD has evaluated this methodology in various countries, demonstrating its validity and consistency [51]. The WHO [52], as well as the OECD [53] encourage its use. (3) The age range considered for the demand indicators of MH services was also 15–19 years. After reviewing a variety of data sources, such as the Birth Information Subsystem, we decided to use data from the National Survey of Demographic Dynamics (*ENADID*). We arrived at this decision because the Birth Information database omits deceased children and abortions. Information from the *ENADID* thus provided the closest approximation to the adolescent pregnancy phenomenon under study. In addition, it has been documented that pregnant women under 15 register their children with the appropriate authorities later than other women [54]. (4) Finally, it should be noted that expenditure was analyzed at the state level, without considering the wide variability in the distribution of local spending [17,50]. Future studies need to explore in greater detail the relationship among MHFP resources, as well as their distribution and health outcomes at the municipal level.

## 5. Conclusions

Governments around the world have recognized the need to invest in the sexual and reproductive health of adolescents. Sustainable Development Goal 3.7 [55] calls upon countries to ensure universal access to sexual and reproductive healthcare services, including for FP. This will require additional resources. Financial evidence on the levels of expenditure allocated to these services and on its distribution throughout the population is a key input for planning public investment. It serves as a basis for governments and health authorities to define how much more they must invest, what types of services should be prioritized, and which areas can be improved with regard to equity and efficiency.

The findings of this study fill an information gap on the levels of investment in sexual and reproductive health services in Mexico for a group traditionally lacking visibility—the adolescents. Our results demonstrate that the health policies implemented between 2003 and 2015 increased expenditure on the sexual and reproductive health of adolescents without Social Security coverage; in spite of this, however, problems persist in ensuring an equitable distribution of these resources. Looking ahead, the implementation of specific policies for the prevention of adolescent pregnancy will require special attention as the current health reform evolves. It will be necessary to monitor the financial implications of the ensuing changes and their consequences for adolescent health services. Subsequent analyses will also need to combine the allocation of expenditure with results indicators in this population in order to understand the extent to which investments are provided equitably and efficiently.

## Figures and Tables

**Figure 1 ijerph-17-03097-f001:**
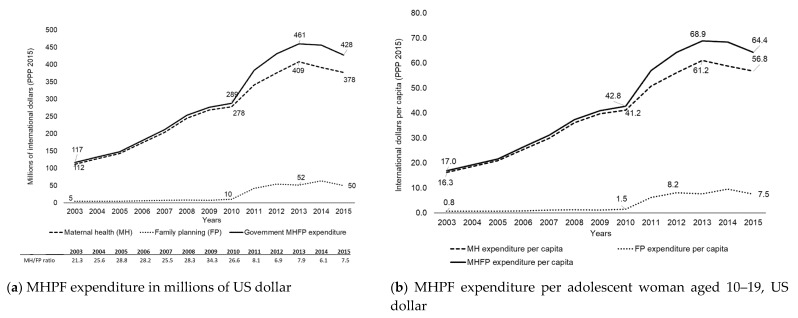
Government Maternal Health and Family Planning (MHFP) expenditure for adolescents aged 10–19 without Social Security coverage, Mexico, 2003–2015, US$ (PPP 2015).

**Figure 2 ijerph-17-03097-f002:**
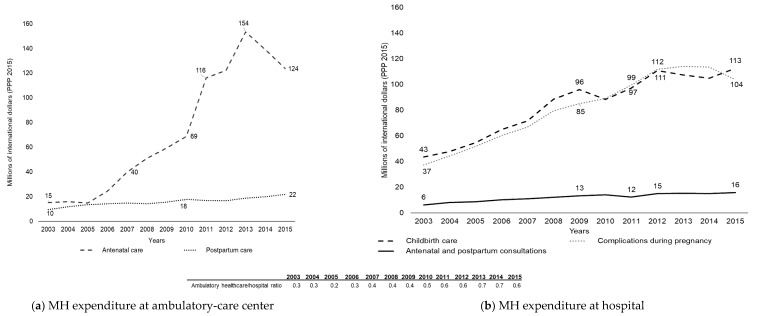
Government Maternal Health (MH) expenditure for adolescent women aged 10–19, without Social Security coverage, by type of healthcare provider, Mexico, 2003–2015, US$ million (PPP 2015).

**Figure 3 ijerph-17-03097-f003:**
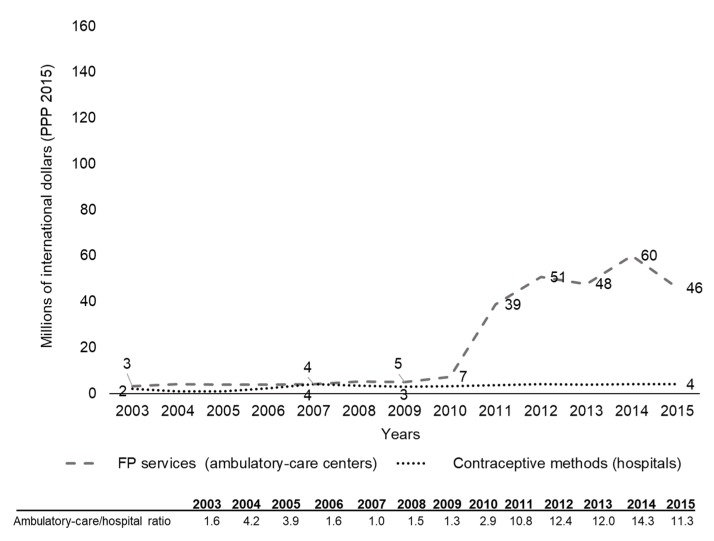
Government Family Planning (FP) expenditure for the entire adolescent population aged 10–19 without Social Security coverage, by type of healthcare provider, Mexico, 2003–2015, $ million (PPP 2015).

**Figure 4 ijerph-17-03097-f004:**
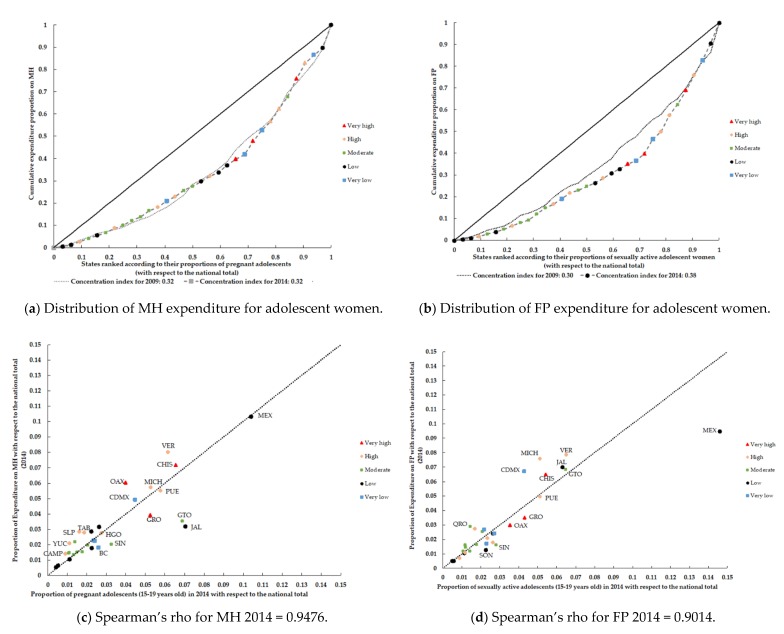
Concentration curves and correlations: government MH and FP expenditure for adolescent women (15–19 years old) without Social Security coverage but with potential demand indicators (Mexico, 2009 and 2014).

**Table 1 ijerph-17-03097-t001:** Average annual percent change in government Maternal Health and Family Planning (MHFP) expenditure for the entire adolescent population aged 10–19 without Social Security coverage, Mexico, 2003–2015.

	2003–2009 AAPC ^a^ [CI 95%]	2009–2011 AAPC ^a^ [CI 95%]	2011–2015 AAPC ^a^ [CI 95%]
Government MHFP expenditure for adolescents	15.4 ^b^ [14.3–16.5]	15.4 ^b^ [14.3–16.5]	3.9 [−3.1−11.4]
Government MH expenditure for adolescent women	16.1 ^b^ [14.4–17.7]	11.6 ^b^ [7.7–15.8]	3.2 [−2.2−8.8]
*Ambulatory-care centers*			
Antenatal care	29.3 ^b^ [23.5–35.3]	29.3 ^b^ [23.5–35.3]	6.1 [−17.4−36.4]
Postpartum care	9.7 ^b^ [3.5–16.3]	5.5 ^b^ [4.2–6.8]	5.5 ^b^ [4.2–6.8]
*Hospitals*			
Complications during pregnancy, childbirth and puerperium	14.3 ^b^ [13.2–15.5]	8.3 ^b^ [5.6–11.0]	1.3 −3.0−5.8]
Childbirth care	14.3 ^b^ [11.9–16.8]	2.7 ^b^ [0.3–5.2]	2.7 ^b^ [0.3–5.2]
Antenatal and postpartum consultations	8.1 ^b^ [4.0–12.5]	3.0 ^b^ [1.6–4.5]	3.0 ^b^ [1.6–4.5]
Government FP expenditure for the entire adolescent population	9.3 ^b^ [2.2–16.9]	92.3 [-8.4-303.9]	16.5 [−7.3−46.5]
*Ambulatory-care centers*			
FP consultations and counseling	4.6 [-0.4-9.8]	130.6 ^b^ [25.5–323.7]	19.1 [−0.5−42.5]
*Hospitals*			
Definitive FP methods	6.6 ^b^ [3.8–9.5]	6.6 ^b^ [3.8–9.5]	6.6 ^b^ [3.8–9.5]

^a^ AAPC: Jointpoint models adjusted for autocorrelated errors due to inertial spending behavior (95%). ^b^
*p* < 0.05 value.

**Table 2 ijerph-17-03097-t002:** Government MHFP expenditure per adolescent woman aged 10–19. States grouped by level of marginalization, Mexico, 2003–2015 ($ [PPP 2015]).

	2003	2004	2005	2006	2007	2008	2009	2010	2011	2012	2013	2014	2015	AAPC ^a^ [CI95%]
**National spending**	17.0	19.3	21.7	26.4	31.1	37.5	41.0	42.8	57.1	64.3	68.9	68.5	64.4	11.8 [9.2–14.5]
Min (state)	7.0	6.5	8.9	11.6	19.8	22.9	26.0	26.3	37.1	38.2	44.8	44.2	38.0	17.2 [10.8–23.9]
Max (state)	42.0	48.9	61.7	84.3	96.6	112.2	106.1	88.9	114.8	158.4	113.8	125.4	119.7	10.1 [4.8–15.7]
Ratio Max/Min	5.96	7.55	6.91	7.25	4.89	4.91	4.08	3.38	3.09	4.15	2.54	2.84	3.15	
**Marginalization ^b^**														
Very low	98	116	123	151	169	229	244	233	274	362	327	306	326	10.0 [6.5–13.6]
Low	198	236	226	289	338	393	460	451	520	569	592	595	572	9.6 [7.2–12.1]
Moderate	193	227	224	274	315	350	395	418	567	657	680	736	693	12.8 [11.3–14.4]
High	160	197	242	308	334	379	386	368	549	582	637	649	616	12.1 [9.8–14.4]
Very high	51	55	62	73	87	104	109	119	159	186	213	212	194	12.6 [9.5–15.8]
Ratio: very high/very low	1.92	2.09	1.98	2.07	1.95	2.19	2.24	1.96	1.72	1.95	1.54	1.44	1.68	

^a^ AAPC: (Average Annual Percent Change); Jointpoint models adjusted for autocorrelated errors due to inertial spending behavior (95%) ^b^ States grouped by level of marginalization [31]: very high marginalization: Chiapas (CHIS), Oaxaca (OAX) and Guerrero (GRO); high marginalization: Puebla (PUE), Hidalgo (HGO), Tabasco (TAB), Veracruz (VER), San Luis Potosi (SLP), Campeche (CAMP), Michoacan (MICH) and Yucatan (YUC); moderate marginalization: Queretaro (QRO), Guanajuato (GTO), Tlaxcala (TLAX), Quintana Roo (QROO), Morelos (MOR), Sinaloa (SIN), Zacatecas (ZAC), Durango (DGO) and Nayarit (NAY); Low marginalization: Jalisco (JAL), Mexico (MEX), Aguascalientes (AGS), Sonora (SON), Colima (COL), Chihuahua (CHIH), Tamaulipas (TAMP) and Baja California Sur (BCS); very low marginalization: Baja California (BC), Coahuila (COAH) Nuevo Leon (NL), and Mexico City (CDMX).

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
