# Peer review of "Government Expenditure on Maternal Health and Family Planning Services for Adolescents in Mexico, 2003–2015"

_ijerph, 2020, doi:10.3390/ijerph17093097_

Round 1
Reviewer 1 Report
Avila-Burgos et al. propose an interesting study on the government expenditure on maternal health and family planning services for adolescents in Mexico using public health data and jointpoint regression. The introduction is extremely well written and informative on the Mexican healthcare system. The discussion and conclusion are rather clear. However, some methodological questions need precisions and answers:
- Why the “demand indicators” (paragraph 2.2.) focus on women? For maternal health it is understandable but it would be interesting to know whether men use FP services? Is it because their level of FP consumption insignificant compared to women?
- Similarly, when identifying and computing the MHFP expenditure for adolescents were men included in the count?
- Overall it is sometimes unclear in the tables, figures, and discussion whether the authors refer to the women or the all adolescent population. Clarification on the status of male adolescents would be interesting to make, in the discussion at least.
- For Figure 4(a) and Figure4(b), it is not clear why the authors label the y-axis as the cumulative expenditure proportion on MH or FP? Why cumulative? In addition, are the distribution curves the result of smoothing? It would be interesting to visualised the actual data points.
- For Figure 4(c) and Figure4(d), it would be interesting to visualise the 5 marginalisation groups using different shading of colours or different type of dots (dots, stars, triangle…)
Reviewer 2 Report
Enumeration, descriptive paper - fine as is, but would have liked to see more discussion and recommendations.
My biggest issue is that the paper was largely descriptive - it summarized what the Mexican government had done to increase access to maternity/birth control services among adolescents. Not included was any sense as to what this had accomplished. Was the perinatal death rate falling among these young women? Were there fewer deaths from complications of illegal abortions? It read like a government-commissioned report on activities rather than a paper aimed at public health professionals.Author Response
Please see the attachmennt.

Round 2
Reviewer 1 Report
Dear authors,
Thank you for your answers. The reading of your paper is improved.
Our answer to my first question was especially clear and some elements of that answer could again enriched the introduction or discussion of your paper.
I would also point out a minor issue. In lines 263-265, I believe you meant Figure 4c and not 4a as the countries' abbreviations are not visible in Figure4a.
